# Enhancing the Viability of a Small Giant Panda Population Through Individual Introduction From a Larger Conspecific Group: A Scientific Simulation Study

**DOI:** 10.3390/ani14162345

**Published:** 2024-08-14

**Authors:** Yuzhen Zhang, Jiabin Liu, Jiaojiao Yu, Cheng Li, Xing Zhao, Li Mo, Wei Wu, Yulin Gai, Qiang Xu, Jiubin Ni, Limin Shen, Haijun Gu, Jindong Zhang, Dunwu Qi, Xiaodong Gu

**Affiliations:** 1Key Laboratory of Southwest China Wildlife Resources Conservation Ministry of Education, China West Normal University, Nanchong 637001, China; zhangyz9711@163.com (Y.Z.); zukoxi98@163.com (X.Z.); yulingai0818@163.com (Y.G.); 2Sichuan Key Laboratory of Conservation Biology for Endangered Wildlife, Chengdu Research Base of Giant Panda Breeding, Chengdu 610081, China; jiabin_liu2013@126.com (J.L.); yujiaojiao2016@126.com (J.Y.); lslicheng@126.com (C.L.); jasmineae@126.com (L.M.); happyeffiewu@163.com (W.W.); 3World Wide Fund for Nature, China Office, Beijing 100006, China; qxu@wwfchina.org; 4The Nature Conservancy (USA) Beijing Representative Office, Beijing 100600, China; nijiubin@163.com; 5Tangjiahe National Nature Reserve, Guangyuan 628100, China; tjhshenlm@163.com; 6Sichuan Forestry and Grassland Bureau, Chengdu 610082, China; ghyciom@163.com (H.G.); gu1751@cn.com (X.G.)

**Keywords:** conservation of giant pandas (*Aliuropoda melanoleuca*), genetic diversity, population rejuvenation, population viability

## Abstract

**Simple Summary:**

Individual quantity is the guarantee of the long-term survival of a population. Currently, approximately 70% of the 33 local populations of wild giant pandas (*Aliuropoda melanoleuca*) face the risk of extinction. We studied four typical populations, from Tangjiahe, Wanglang, Liziping, and Daxiangling, and attempted to translocate individuals from large populations to enhance the viability of small populations. Our results validated a classic conclusion that the introduction of individuals from Tangjiahe and Wanglang significantly improved the genetic diversity in Liziping and Daxiangling. To ensure the long-term survival of two small populations over the next 100 years, we have devised multiple specific strategies for individual introduction. Our research has certain value for the rejuvenation of small populations of giant pandas, and we hope to provide a reference for the protection of endangered animals such as giant pandas.

**Abstract:**

Currently, nearly 70% of giant panda populations are facing survival challenges. The introduction of wild individuals can bring vitality to them. To explore this possibility, we hypothetically introduced giant pandas from Tangjiahe and Wanglang into Liziping and Daxiangling Nature Reserves. We collected feces from these areas and analyzed the genetic diversity and population viability before and after introduction using nine microsatellite loci. The results showed the genetic level and viability of the large populations were better than the small populations. We investigated the effects of time intervals (2a, 5a, and 10a; year: a) and gender combinations (female: F; male: M) on the rejuvenation of small populations. Finally, five introduction plans (1F/2a, 2F/5a, 1F1M/5a, 3F/10a, and 2F1M/10a) were obtained to make Liziping meet the long-term survival standard after 100 years, and six plans (1F/2a, 2F/5a, 1F1M/5a, 4F/10a, 3F1M/10a, and 2F2M/10a) were obtained in Daxiangling. The more females were introduced, the greater the impact on the large populations. After introducing individuals, the number of alleles and expected heterozygosity of the Liziping population are at least 6.667 and 0.688, and for the Daxiangling population, they are 7.111 and 0.734, respectively. Our study provides theoretical support for the translocation of giant pandas, a reference for the restoration of other endangered species worldwide.

## 1. Introduction

More than 157,100 species are currently recorded in the IUCN Red List, of which over 44,000 species are facing the threat of extinction [1]. Biodiversity is crucial for nature conservation [2]. The preservation of endangered species and the achievement of sustainable species development have long been the central focus and primary research direction for conservation biologists. As a rare relict species in China, giant pandas (*Aliuropoda melanoleuca*) serve as flagship and umbrella species [3]. The conservation efforts dedicated to giant pandas not only offer protection to other species inhabiting the same region but also provide valuable insights for safeguarding other endangered species worldwide [4]. In recent years, there has been an overall increase in the population of giant pandas, leading to a decrease in their threatened status from endangered to vulnerable [5]. However, the survival prospects for small populations are not very optimistic due to habitat fragmentation and increased human activities [6,7].

In order to reduce the damage of human activities and habitat fragmentation to the giant panda population and better protect them in situ, China has established 67 giant panda nature reserves. However, despite these measures, nearly 33% of the population still remains unprotected [8]. Over the past decade or so, increasing the population size through individual releases has been a major focus of researchers [9]. Looking back at the releases, the survival rates of about 75% for wild individuals released, the survival rates of released captive individuals with wild lineage is about 50%, and no clear evidence of survival after the release of captive individuals of captive parentage have obtained. Of this total, about 66.7% of the individuals had wild individuals as their parents. As the most successful case of wild individual release, “Luxin” is the only panda found to have given birth to a cub in the wild [10,11,12]. Individual introduction can reduce the risk of extinction of small populations [13], and historical data suggests that ex situ releases of wild individuals have the highest success rates. In contrast to individuals which often led to the development of behavioral defects due to the captive environment [14], researchers would not have to devote considerable time and effort to develop the survival skills of wild individuals [15]. Although the establishment of ecological corridors can also provide assistance [16], the implementation of these measures may face constraints in terms of financial resources and time availability, and there is no guarantee that they will be fully utilized by pandas [17]. Therefore, directly using wild individuals as a source of introduction is an ideal approach for the genetic rescue of small populations [18], and this will also be a new perspective on small population restoration.

With the idea that the giant pandas have reached an evolutionary dead end refuted, it is widely accepted that the giant panda is a species with a medium or high genetic diversity [19,20,21]; the evolutionary potential is not low compared to species in the same family [22]. However, looking at the six mountains, the Xiangling areas were comparatively lower [23]. Under natural conditions, populations with lower genetic levels often face a greater risk of extinction and show a decreased resistance to random factors [24,25]. Population viability analysis, through simulating population dynamics, concretizes the challenges that giant pandas may face at a specific point in the future [26]. Although population parameters vary slightly among similar studies, one consistent finding is that smaller populations tend to exhibit weaker viability [27,28,29]. From past research, most studies have focused on the population of giant pandas in entire mountains or counties [22,30,31]. Due to the vast study areas, these investigations may fail to fully capture the genetic dynamics and survival status of the smaller, localized areas. The Tangjiahe and Wanglang populations belong to the large population in the Minshan Mountains with good habitat connectivity, while the Liziping and Daxiangling populations are distributed in the Xiangling Mountains with broken habitats [32]. Our research selected the four populations as the study objects and conducted a comprehensive assessment of the genetic status and viability of giant panda populations. Utilizing the findings from the studies, we simulated the introduction of individuals from a population with a high genetic diversity to address the deficiencies observed in the small populations. This is the first theoretical study on the introduction of wild individuals. Our plan aims to mitigate the risks associated with genetic bottlenecking and enhance the long-term survival prospects of these populations. Furthermore, it is our aspiration that this study can provide valuable insights and guidance for the conservation efforts dedicated to giant pandas, as well as other endangered species worldwide.

## 2. Materials and Methods

### 2.1. Study Area and Sample Collection

Our study area includes four nature reserves: Tangjiahe (TJH), Wanglang (WL), Liziping (LZP), and Daxiangling (DXL), which are respectively located in Qingchuan, Pingwu, Shimian, and Yingjing Counties in Sichuan Province, Southwest China. In the fourth survey conducted from 2011 to 2014, 39 wild giant pandas were found in TJH, 28 in WL, 22 in LZP, and seven in DXL. From 2019 to 2023, we investigated the number of giant pandas in these four regions again and 156 fecal samples were collected from TJH, 126 from WL, 92 from LZP, and 40 from DXL (Figure 1). All samples were collected with sterile gloves and stored in more than 95% anhydrous ethanol.

### 2.2. DNA Extraction and Microsatellite Amplification

The total DNA from fecal samples was extracted using the QIAamp Fast DNA Stool Mini Kit (Hilden, Germany) following the manufacturer’s instructions and stored at −20 °C. In this study, nine microsatellite markers with high polymorphic information content and stability were utilized (Table 1). The upstream primer was fluorescently labeled with FAM, HEX, and TAM at the 5’ end. The reaction system was 20 uL: Taq PCR Mastermix 10 μL, upper and downstream primers 0.5 μL each, DNA 2 μL, BSA 1 μL, and ddH_2_O was added to 20 μL. The reaction conditions: 95 °C for 15 min; 94 °C for 30 s, 48 °C~60 °C for 90 s, 72 °C for 60 s, for a total of 30 cycles; 60 °C for 30 min; and stored at 4 °C. The PCR products were detected through electrophoresis on a 2% agarose gel. Primer synthesis and microsatellite genotyping were conducted by Sangon Biotech Co., Ltd. (Shanghai, China).

### 2.3. Data Analysis

#### 2.3.1. Individual Identification

The Micro-Checker 2.2.3 was utilized to identify any missing or invalid alleles [33]. The Gimlet 1.3.3 software was employed to calculate the *p*-value for the joint differentiation rate among the 9 loci [34]. Subsequently, the Microsatellite Toolkit was employed to discern individual similarities [35]. Samples were considered to belong to the same individual if either all alleles across the nine loci were identical or only one allele within a single locus differed.

#### 2.3.2. Genetic Diversity and Genetic Structure

GenAlEx 6.5 was used to calculate the number of alleles (Na), observed heterozygosity (Ho), expected heterozygosity (He), and Shannon–Wiener Index (I) [36]; the Hardy–Weinberg equilibrium was tested by Genepop 4.7 [37]. 

Structure 2.3.4 was used to analyze the source of individuals in a population by the Bayesian clustering method [38]. The value of K was set to range from one to eight, and ten independent operations were carried out. Before 1,000,000 formal calculations were repeated, 100,000 preliminary experiments were conducted. The calculation results were estimated using the Structure Harvester online tool to determine the optimal K value [39].

#### 2.3.3. Population Viability

The extinction rates of four populations in the next 100 years were calculated using Vortex 10.6 [40], and each simulation was repeated 1000 times. Parameter setting: The initial breeding age of female and male wild giant pandas is seven and eight years old, respectively, the maximum breeding age is 20 years old, and the maximum lifespan is 26 years old [41,42]. The sex ratio is 1:1, and the annual reproductive rate is 62.5%, with a single birth rate of 58.33% and a twin birth rate of 4.17%, and all males have the ability to engage in reproduction [41]. Additionally, the mortality rates at different age stages can be found in Appendix A [41]. Natural disasters include bamboo flowering and forest fire, the frequency of bamboo flowering is 1/60, and the frequency of forest fire is 1/30; their impact on survival and reproduction is both 10% [43,44]. The lethal equivalent of inbreeding depression is 3.14 based on a study of 40 captive mammals in North American zoos [45]. The criteria for long-term survival are an extinction rate of less than 2% and gene diversity greater than 0.9 [46].

The number of individuals in the reserve obtained from this study represents the initial population size. The maximum population density of giant pandas was 3.03/km^2^ [47,48]. The area of the reserves and the coverage rate of edible bamboo for giant pandas are sourced from the fourth survey report [32]. The maximum environmental capacity can be found in Table 2.

## 3. Results

### 3.1. Genotyping and Individual Identification

A total of 227 reliable genotypes were identified from the samples. Micro-checker analysis indicated that the amplification results were not affected by null alleles or allele dropout. The joint differentiation rate of nine loci was high, with a PID value of 1.73 × 10^−9^, and the probability P (sib) of misjudgment caused by twins was 4.06 × 10^−4^. Even if the locus with the highest polymorphism (GPL-47) fails to amplify, the probability of misjudgment caused by twins only increases to 1.15 × 10^−3^, which is much smaller than 0.01 [51]; this satisfies the requirement for population size evaluation. Therefore, this study retained genotypes at eight loci and obtained a total of 139 unique genotypes. Among them, there are 56 genotypes from TJH, 45 from WL, 25 from LZP, and 13 from DXL.

### 3.2. Genetic Diversity and Genetic Structure

A total of 83 alleles were detected from the 139 unique genotypes. There were 17 private alleles that were present only in a single population, and six of these occurred in the TJH, five in the DXL, and three in the WL and LZP, respectively. Only 66 alleles were shared among all four populations.

The average allele numbers were 7.556 and 6.333 for TJH and WL populations, respectively, while LZP and DXL populations had average allele numbers of 5.667 and 5.778, respectively. The expected heterozygosity (He) ranked from high to low were 0.746 (WL), 0.743 (DXL), 0.725 (TJH), and 0.654 (LZP). The highest observed heterozygosity was 0.578 (WL), followed by 0.536 (TJH) and 0.530 (LZP), and the lowest was 0.430 (DXL). The Shannon–Wiener values ranked from high to low were 1.534 (TJH), 1.531 (DXL), 1.528 (WL), and 1.316 (LZP). The comprehensive genetic parameters indicated that the genetic level of TJH and WL populations was higher, followed by DXL and LZP populations. The Hardy–Weinberg equilibrium test revealed that there were seven loci with deviations from equilibrium in TJH, six in WL and DXL, and four in LZP (*p* < 0.01) (Table 3). 

Then, we analyzed the structure of the four populations individually. The TJH and WL populations both exhibit three gene clusters, suggesting that they originate from three subpopulations. The DXL and LZP populations are composed of two gene clusters, indicating that they originate from two subpopulations (Figure 2 and Figure 3).

### 3.3. Population Viability

In an ideal state without inbreeding depression and natural disasters such as bamboo flowering and forest fires, the extinction rates of the four populations were 2.4% (TJH), 4.0% (WL), 19.6% (LZP), and 52.5% (DXL), respectively, and none of them meet the requirements for long-term survival. The initial number of individuals is synergistically related to genetic diversity and inversely proportional to the cumulative extinction rate. The TJH (N = 56) population has the strongest viability, while the DXL (N = 13) population has the highest risk of extinction. Under the separate effects of inbreeding depression or natural disasters, the former has a stronger negative impact on the LZP and DXL populations compared to the latter. If the two kinds of effects coexist, the extinction rate of the four populations will increase significantly, and the extinction rate of small populations is higher than 50%, which is about three~six times that of large populations. It is important to note that there is the migration of exotic individuals among the TJH and WL populations. Therefore, it is possible that the defined extinction risk is relatively high based on a set of static values. The two small populations on the brink of extinction are relatively isolated, and if no assistance is provided by humans, it is highly likely that they will vanish within the next 100 years (Figure 4).

### 3.4. Rejuvenation of Small Populations

#### 3.4.1. Viability of Small Populations after Introducing Individuals

This study explored the effects of introducing individuals with different gender combinations (female: F; male: M) every two, five, and ten years (year: a) on the viability of small populations under ideal conditions. When introducing individuals, start increasing from one only until long-term survival conditions are met (Appendix A).

We designed 16 plans to rejuvenate LZP and ultimately found that five plans were the most suitable (Figure 5). The extinction rate of all the applicable plans was zero, so they are not shown in the figure. Based on the extinction rate, genetic diversity, and the expected number of individuals in the next 100 years, it is most beneficial for the survival of LZP to introduce 1F/2a, then 2F/5a and 1F1M/5a, and finally 3F/10a and 2F1M/10a. More females means a higher birth rate, which can produce more new individuals. Therefore, the growth rate of individual numbers in the LZP population was the highest when 1F/2a was introduced, while the growth rate was lowest when 1F1M/5a or 2F1M/10a were introduced.

For the DXL with the smallest initial number of individuals, we explored 21 plans, of which only six plans were applicable (Figure 5), and the extinction rate was zero. The most significant way to expand the population of DXL is still to replenish 1F/2a. Then, there are 1F1M/5a, 3F1M/10a, and 2F2M/10a, which have similar effects. Finally, there are 2F/5a and 4F/10a, which have weaker enhancement effects on the viability of the DXL population compared to the first four plans. As for the population size of DXL after 100 years, it still depends on the number of females that were introduced.

#### 3.4.2. Viability of Large Populations after Introducing Individuals

We further investigated the impact of applicable introduction plans on two large populations (Figure 6; Appendix A). Among the five plans applicable to LZP, 1F/2a has the greatest impact, which leads to the highest extinction rate of TJH and WL in the next 100 years, and the lowest gene diversity and individual number. Then, the damage to the large populations in descending order is 2F/5a, 3F/10a, 1F1M/5a, and 2F1M/10a. The more females were introduced, the smaller the population size of TJH and WL in 100 years.

Among the six plans applicable to DXL (Figure 6), 1F/2a is still the most harmful introduction plan for TJH and WL. The impact of 2F/5a and 4F/10a on the large population is slightly lower than that of 1F/2a. Introducing 3F1M/10a can obviously show that the gap between the large populations and their ideal state has narrowed. In addition, 1F1M/5a and 2F2M/10a are the least harmful to large populations.

#### 3.4.3. Genetic Level of Small Populations after Introducing Individuals

##### Estimation of Genetic Level of LZP

In an ideal state, the number of giant pandas in TJH after 100 years is 113 (Appendix A). After introducing individuals to LZP according to five introduction plans, The number of remaining individuals in TJH was 28 (introduce 1F/2a)~71 (introduce 2F1M/10a), with the introduced rate of 37% [(113 − 71)/113]~75% [(113 − 28)/113]. By introducing 21 (56 × 37%)~42 (56 × 75%) individuals from 56 TJH individuals to LZP, the genetic level of LZP after 100 years was estimated (Figure 7A). Selecting the 42 individuals of TJH with the highest genotype similarity to LZP individuals, when introducing the top 21 individuals, the number of alleles and expected heterozygosity were 7.111 and 0.688, respectively. When all were introduced, the number of alleles and expected heterozygosity were 7.667 and 0.703, respectively. Therefore, it is speculated that the number of alleles of LZP after introducing individuals from TJH will be 7.111 to 7.667, and the expected heterozygosity will be 0.688 to 0.703.

The number of giant pandas in WL after 100 years is 90 in an ideal state. After revitalizing LZP, the number of remaining individuals in WL was about 17 (introduce 1F/2a)~56 (introduce 2F1M/10a), and the introduced rate was 38% [(90 − 56)/90]~81% [(90 − 17)/90]. The genetic status of LZP in the future can be estimated by introducing 17 (45 × 38%)~36 (45 × 81%) individuals from the current 45 WL individuals (Figure 7B). The number of alleles and expected heterozygosity of LZP were 6.667 and 0.706, respectively, when 17 individuals of WL with the highest similarity to LZP individuals were selected. When 36 individuals were selected, the number of alleles and expected heterozygosity were 7.000 and 0.741, respectively. Therefore, it is speculated that when introducing individuals from WL to LZP, the number of alleles in LZP will be 6.667~7.000, with an expected heterozygosity of 0.706~0.741.

##### Estimation of Genetic Level of DXL

Similarly, for the rejuvenation of DXL (Appendix A), the introduced rate of TJH is 38% (introduce 1F1M/5a; [(113 − 70)/113])~75% (introduce 1F/2a; [(113 − 28)/113]), and it is estimated that in the future, the genetic level of DXL will need to introduce 21 (56 × 38%)~42 (56 × 75%) individuals from TJH (Figure 7A). The number of alleles and expected heterozygosity were 7.556 and 0.734 when introducing the top 21 individuals, respectively. When introducing 42 individuals, the number of alleles and expected heterozygosity were 7.889 and 0.740, respectively. Therefore, it is speculated that the number of alleles in DXL will be 7.556~7.889 and the expected heterozygosity will be 0.734~0.740 when individuals are introduced from TJH to DXL.

For rejuvenating DXL, the introduced rate of WL is 39% (introduce 1F1M/5a; [(90 − 55)/90])~81% (introduce 1F/2a; [(90 − 17)/90]). According to this ratio, WL needs to introduce 18 (45 × 39%)~36 (45 × 81%) individuals to speculate the genetic level of DXL (Figure 7B). When the first 18 individuals with the highest similarity to DXL were introduced, the number of alleles and expected heterozygosity were 7.111 and 0.758, respectively. When 36 individuals were introduced, the maximum number of alleles and expected heterozygosity were 7.333 and 0.764, respectively. Therefore, it is speculated that the number of alleles of DXL will be 7.111~7.333, and the expected heterozygosity will be 0.758~0.764 when individuals are introduced from WL to DXL.

**Figure 7 animals-14-02345-f007:**
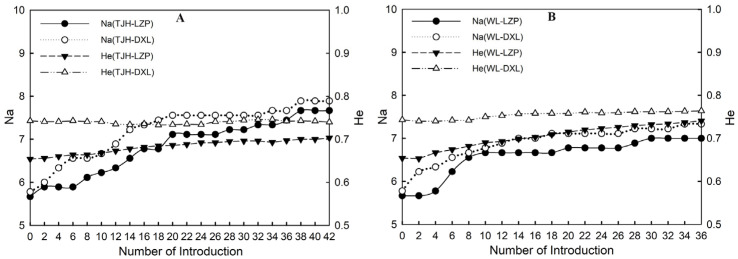
The future genetic level of LZP and DXL populations. Note: (**A**) shows the genetic level changes of two small populations after introducing individuals from TJH; (**B**) shows the genetic level changes of two small populations after introducing individuals from WL.

## 4. Discussion

Reserves are highly effective tools for biodiversity conservation, and their establishment and systematic planning represent a valuable legacy that can be passed down through generations in the field of conservation [52]. TJH and WL Nature Reserves are part of population A, situated in the northern region of the Minshan Mountains, where the largest known population of wild giant pandas currently exists [53]. The population of giant pandas in TJH inhabits Qingchuan County and maintains communication with the wild populations in Wen County and Pingwu County [54]. Since the fourth survey, our research has found that the population of TJH has increased to 56 individuals, a growth of approximately 43.6% [32]. As a core part of the Jiuzhai–Baima local population, the population of WL has increased by nearly 60.7% compared to the previous survey, with a current count of around 45 individuals [32], due to the healthy gene exchange among large populations. Of the existing wild populations in the six mountains, the Xiaoxiangling populations were geographically separated by the presence of a railway and the Dadu River, impeding gene exchange with other mountain populations. Consequently, this isolation poses an extremely high risk of extinction [55,56]. In particular, the LZP Nature Reserve in Shimian County, which is divided into two parts by National Highway 108, serves as a prime example of isolated small populations. Daxiangling is the mountain with the lowest population density of giant pandas, except for Xiaoxiangling [8]. The distribution of individuals is also quite scattered, with the majority living in Yingjing County [57]. In recent years, the population growth in DXL Nature Reserve has been less than ten individuals [32], and the severity of inbreeding depression is a cause for concern [58]. In macro perspective, the habitat research of giant pandas has been quite comprehensive [59], while the scientific questions at the micro level still intrigue us. Specifically, we are curious about the genetic differences and survival abilities among wild populations of different scales, as well as the most favorable approach for individual introduction to rejuvenate small populations. The research objects we have chosen are relatively representative and can provide us with an objective answer.

Genetic variation reflects the evolutionary potential of species [60]. This study employed nine microsatellite markers, with polymorphic information content ranging from 0.648 (Panda-22) to 0.810 (GPL-47), all of which were higher than 0.5, this indicates that these markers can provide ample genetic information for our research [61]. The TJH and WL populations have a high number of alleles and heterozygosity, indicating that their genetic diversity is at a high level, which is consistent with previous research findings [62,63]. In addition to the geographical advantage of the reserves, the efforts made by the government in management and protection should not be ignored [18]. The small populations have relatively lower levels of genetic diversity and genetic structure analysis showed that they also have fewer ancestral components. We further adopted population viability analysis and used previous wild giant panda parameters to evaluate the long-term survival ability of the four populations [41,44,46,64], and obtained the same results. In an ideal state, the extinction rate of DXL is as high as 52.5%, while LZP is 19.6%. Their situation is much more dangerous compared to TJH’s 2.4% and WL’s 4.0%. It is also not difficult to observe from the results that small populations are more susceptible to the effects of inbreeding depression. If inbreeding depression and natural disasters occur together, the extinction rates of LZP and DXL will be increased to 57.9% and 86.7%, respectively, which is about three~six times that of the two large populations at this time. Essentially, this is a matter of population size [65]. The greater the initial number of individuals, the higher the probability of survival [66]. If the number of individuals in the four populations is doubled in the ideal state, the two large populations far exceed the long-term survival standard, and the extinction rate of DXL is reduced by about three times and LZP by about ten times. Compared to the rejuvenation of isolated small populations, increasing the number of large populations requires less effort. We believe that it is expected to realize the self-maintenance of wild populations in the near future by making large populations stronger first and then providing provenances for small populations.

Based on this, we explored the number and sex of introductions needed to meet the long-term survival criteria for the small populations at intervals of two, five, and ten years in an ideal state. Introducing 1F/2a, 2F/5a, and 1F1M/5a is sufficient for DXL and LZP. If replenishment is made every ten years, the DXL with fewer initial individuals needs four individuals to meet the standard, while the LZP needs only three individuals. Obviously, populations with fewer individuals always require more attention and assistance [67]. Moreover, an interesting finding is that in the DXL with the smallest population size, the extinction rates are all zero, and the gene diversity obtained from 1F1M/5a, 3F1M/10a, and 2F2M/10a are slightly higher than that obtained from 2F/5a and 4F/10a. Aligning with the perspectives of Yang et al. [68], the female-biased sex combinations provide greater benefits to population survival compared to introducing only one sex or having more males. Certainly, the giant panda is a species with female-biased dispersal, it has a polygynous mating mechanism [69], and the population litter size is determined by the number of females [54]. Therefore, as far as the plans we have designed are concerned, the more females were introduced to DXL and LZP, the larger the estimated population size will be in 100 years [70]. On the contrary, when these individuals come from TJH or WL, the more females were introduced, the lower the survival rate of TJH and WL. In particular, the most effective introduction plan 1F/2a for LZP and DXL made the extinction rate of TJH soar from 2.4% to 46.7%, and WL soared from 4.0% to 66.4%. The plans that not only meet the long-term survival of small populations but also have less harm to large populations are 1F1M/5a, 2F1M/10a, and 2F2M/10a. Thus, fewer males can be taken into account in our selection of individuals.

Previous research has shown that after the introduction of individuals, there are new gene frequencies and alleles in this small group [68,71]. We compared the expected number in large populations after 100 years in an ideal state with the number of large populations after introducing individuals into small populations, and we estimated the number of TJH and WL that need to be introduced by reducing the proportion of individuals, so that the genetic level that the small population may reach after 100 years can be inferred. Consistent with previous studies, the genetic level of LZP and DXL will be significantly improved in the future. No matter which large population is selected, the number of alleles of LZP and DXL can reach at least 6.667 and 7.111, respectively, and the expected heterozygosity can reach at least 0.688 and 0.734, respectively. Compared with the genetic level of giant pandas in five mountains except Daxiangling studied by Zhang et al. [22], DXL and LZP populations after rejuvenation have richer genetic diversity. 

We cannot deny that wild individual translocation is the method with the fastest and the highest success rate in saving small populations [18,72]. From a short-term perspective, the introduction of new genetic resources can quickly improve the genetic diversity of small populations [18,71]. From a long-term perspective, migrating individuals from other populations to establish heterogeneous populations reduces the risk of extinction and alters the developmental trajectory of extinction at a certain point in the future [73]. However, as it stands, it is not easy to implement. The direct introduction of wild individuals is not realistic [74], we can only carry out it indirectly, and a better introduction plan may need to be studied in the future. Here are some indirect individual introduction plans, which may provide ideas for the emergence of better schemes. Firstly, the endangerment of giant pandas is linked to the environmental pressures they are under [75]. The Xiangling Mountains present a more challenging environment for the survival of giant pandas [76], it is separated from Qionglai Mountains and Liangshan Mountains by the influence of rivers, railways, and national highways [32]. It is possible to establish ecological corridors to introduce individuals from the large populations of Lewu and Baishahe into the Xiangling Mountains. Secondly, like “Luxin”, individuals rescued from other mountains can be reintroduced into the Xiangling Mountains, thus achieving long-distance individual translocation. In addition, this would allow the offspring of wild individuals rescued to be born and raised in semi-wild conditions before being released into small populations, similar to “Zhangxiang” [70]. It should be noted that based on our existing data, it may not be appropriate for TJH and WL to assist small populations in the current situation where they cannot sustain themselves in the long term. However, it is not difficult to achieve large population growth by strengthening genetic exchange with surrounding populations and protecting habitats [70,77]. A planned individual introduction would be a win–win for both large and small populations at this time. In the future, it is hoped that self-sustainability can be achieved by introducing wild individuals, thereby reducing the pressure on the wild release of captive individuals. Of course, in further research, we need to consider more factors to ensure the successful implementation of the rejuvenation plan, such as the topography of the habitat, the types and areas of edible bamboo, and the interrelations between intruders and residents.

## 5. Conclusions

Currently, the rejuvenation of small populations mainly relies on the release of captive individuals into the wild, which requires significant human, material, and financial resources. Our research indicated that the genetic diversity of large wild populations was higher than that of small isolated populations. It can help small populations overcome their challenges by introducing individuals. Enhancing gene flow between local populations is crucial. It is necessary for us to continue conducting in-depth research on the specific plans for implementing individual introduction in the future.

## Figures and Tables

**Figure 1 animals-14-02345-f001:**
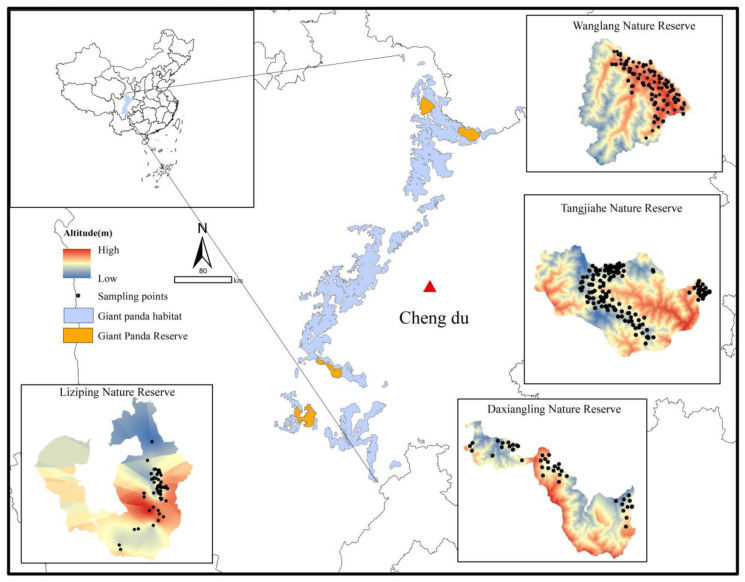
Sampling distribution points of four giant panda nature reserves.

**Figure 2 animals-14-02345-f002:**
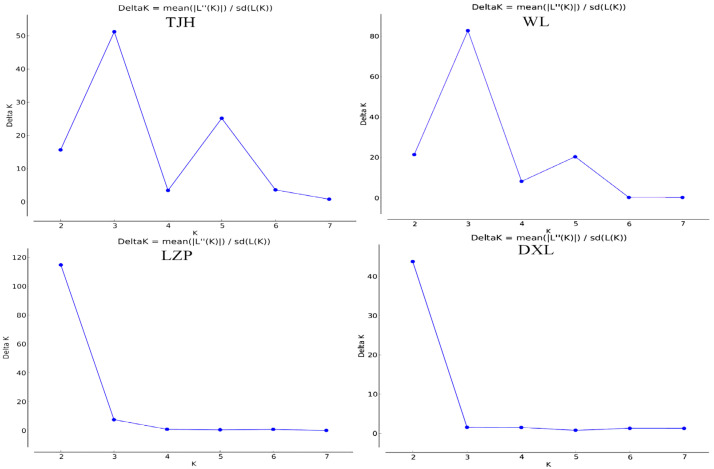
Optimal K values for the analysis of the genetic structure of the four populations.

**Figure 3 animals-14-02345-f003:**
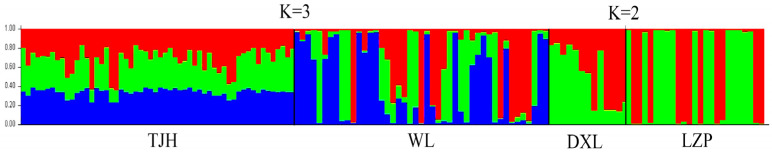
The map of the gene cluster of four giant panda populations. Note: Red, green, and blue represent different gene clusters.

**Figure 4 animals-14-02345-f004:**
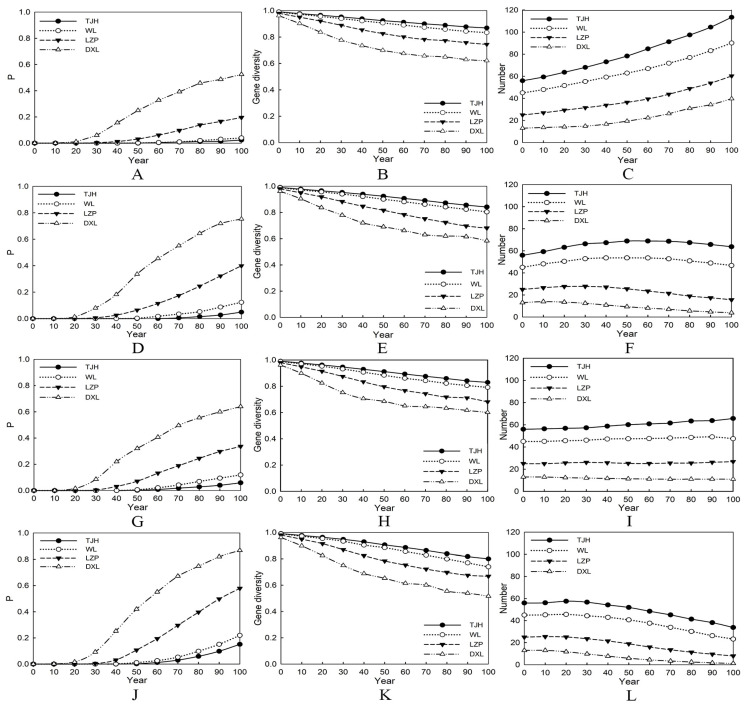
Variation of viability under different conditions in the next 100 years. Note: (**A**–**C**) are unaffected by inbreeding depression and disasters; (**D**–**F**) are affected by inbreeding depression only; (**G**–**I**) are affected by disasters only; (**J**–**L**) are affected by both inbreeding depression and disasters; P is the extinction rate, same as below.

**Figure 5 animals-14-02345-f005:**
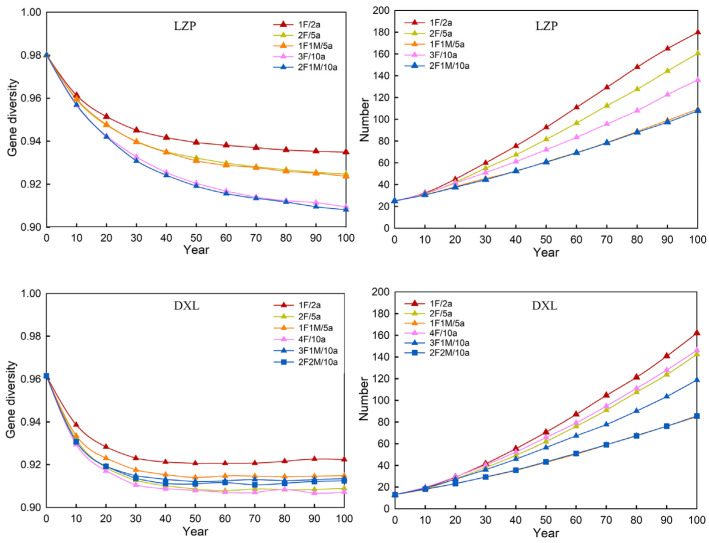
Introduction plans for small populations to meet long-term survival standards. Note: “Year” is represented by “a”; “female” and “male” are represented by “F” and “M”, respectively, same as below.

**Figure 6 animals-14-02345-f006:**
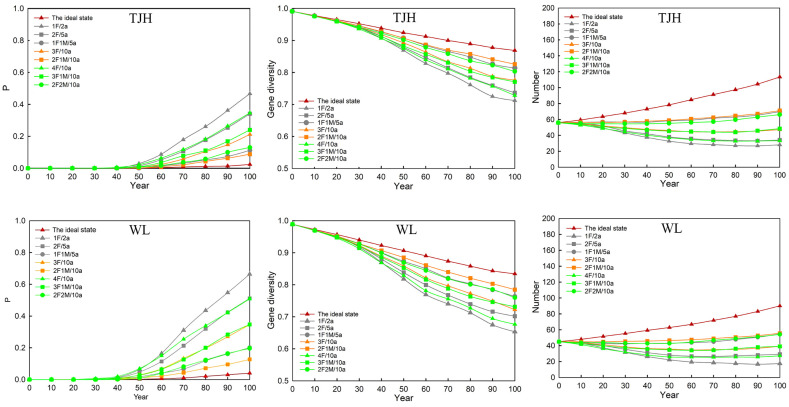
Effect of suitable introduction plans on large populations. Note: The gray is suitable for DXL and LZP, the orange is only suitable for LZP, and the green is only suitable for DXL; the red is the viability of large populations before introduction.

**Table 1 animals-14-02345-t001:** Information of nine microsatellite loci.

Loci	Repeat Motif	Primer Sequence (5′–3′)	Size (bp)
Panda-22	(CAA)12	F: AGGGGAGAGAACATTGCTCG R: GAAGCCAGCCCAACTTTTCC	177–186
Ame-μ26	(CA)11	F: TTTTCAGGCCTCCGAAAAC R: ATTCCCAATAAAGCAAATCAGA	114–120
GPL-60	(TCTT)12	F: TGCCGGAAAGTTCTAAGCAT R: TTTCTCTCCCTCTCCCCTTC	218–238
Ame-μ13	(CA)18	F: GGAAGCATTAAGGAAAACATGC R: AATGATGACCATTTCAAACGC	142–171
Ame-μ11	(CA)12	F: TATGCCACCTGCCCAGAC R: GATGGAAAGAGTAGAGCCAAGG	228–236
Ame-μ10	(CA)16	F: ACCGTGCTCTTAATCCCCTT R: CCCATGCTTATGAGAAACAGG	138–160
GPZ-6	(AAAG)11	F: CCTGGCAGGGCAAAGTATT R: CCCCGTGAAAACATCAAGAC	194–222
GPZ-47	(AATG)20	F: GACCTCAGTGTACGCCCAGT R: CTGGACAGGCAGGTAGAAGC	174–210
GPL-47	(TCTA)20	F: TCCCCCTCTATGGTAAAAGG R: CCATGTTGGGTGTAGGGATT	140–172

**Table 2 animals-14-02345-t002:** Maximum environmental capacity of giant pandas in four nature reserves.

Reserve	Area (km^2^)	Suitable and Sub-Suitable Habitats	Coverage Rate of Edible Bamboo	Maximum Capacity
TJH	400.00	51.00% [49]	83.94%	519
WL	322.97	-	41.81%	409
DXL	284.50	-	98.58%	850
LZP	479.40	26.58% [50]	69.90%	270

**Table 3 animals-14-02345-t003:** Genetic diversity of four populations based on nine loci.

	TJH (n = 56)					WL (n = 45)					LZP (n = 25)					DXL (n = 13)				
	Na	I	Ho	He	P	Na	I	Ho	He	P	Na	I	Ho	He	P	Na	I	Ho	He	P
Panda-22	4.000	1.249	0.547	0.682	0.0303	5.000	1.005	0.778	0.575	0.9990	4.000	1.204	0.250	0.666	0.0001	5.000	1.413	0.769	0.722	0.0486
Ame-μ26	7.000	1.554	0.518	0.748	0	4.000	1.276	0.778	0.704	0.8261	5.000	1.201	0.560	0.594	0.0146	5.000	1.487	0.231	0.749	0
GPL-60	5.000	1.396	0.554	0.715	0.0564	5.000	1.534	0.659	0.767	0.0713	4.000	0.961	0.480	0.554	0.2801	5.000	1.380	0.385	0.710	0.0023
Ame-μ13	11.000	1.589	0.607	0.705	0	7.000	1.347	0.452	0.684	0	6.000	1.220	0.400	0.610	0.0026	9.000	2.034	0.308	0.849	0
Ame-μ11	6.000	1.315	0.518	0.672	0.0005	4.000	1.354	0.682	0.735	0.0899	5.000	1.026	0.640	0.538	0.6727	5.000	1.445	0.333	0.743	0.0007
Ame-μ10	8.000	1.681	0.547	0.771	0	7.000	1.675	0.300	0.787	0	8.000	1.770	0.800	0.801	0.3761	6.000	1.662	0.692	0.790	0.0120
GPZ-6	8.000	1.438	0.464	0.645	0	6.000	1.655	0.659	0.787	0.0057	5.000	1.127	0.320	0.570	0.0045	4.000	1.029	0.308	0.559	0.0086
GPL-47	10.000	1.880	0.618	0.816	0.0001	9.000	1.953	0.548	0.838	0	6.000	1.455	0.800	0.729	0.7751	7.000	1.654	0.462	0.769	0.0270
GPZ-47	9.000	1.704	0.446	0.770	0	10.000	1.955	0.350	0.838	0	8.000	1.880	0.520	0.827	0	6.000	1.667	0.385	0.796	0
Average	7.556	1.534	0.536	0.725	——	6.333	1.528	0.578	0.746	——	5.667	1.316	0.530	0.654	——	5.778	1.531	0.430	0.743	——

Note: The number of individuals (n); the number of alleles (Na); the observed heterozygosity (Ho); the expected heterozygosity (He); Shannon–Wiener index (I); Hardy–Weinberg equilibrium (P).

## Data Availability

The data that have been used are confidential.

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
