# Peer review of "Enhancing the Viability of a Small Giant Panda Population Through Individual Introduction From a Larger Conspecific Group: A Scientific Simulation Study"

_animals, 2024, doi:10.3390/ani14162345_

Round 1
Reviewer 1 Report
Comments and Suggestions for Authors The aim of this study is the analysis of genetic situationin giant panda populations for sustainable conservation
of this species. The study contains also the scientific
simulation of rejuvenation of small populations that could
be applied to avoid the extinction of pandas. I consider the
manuscript to be very valuable for the rational and
sustainable management of panda populations, but it will
also be necessary to verify the conclusions in practical
management.
Fig 4 and 6 will be difficult to distinguish at this magnification
line 210 is duplicated
Line 261 (Table S2 of supplementary material) – where is it?
Table 3 necessary corrections in the title
Author Response
Dear reviewer:
Thank you very much for taking the time to review this manuscript. We appreciate these important comments and will make every effort to revise the manuscript accordingly. We highlighted the revised content in red. The following is a detailed response to the reviewer (see attachment).
Sincerely,
Jindong Zhang

Reviewer 2 Report
Comments and Suggestions for Authors
Short summary: some statements are not new
Introduction: please keep attention that scientific names of species should be italicized; the citations in the text are nor meet MDPI requirements
Some terms are less suitable of the translocation process of wildlife species. Authors should be more careful as due to typos as to using of terms
Some very important publications directly related to species introduction and genetic diversity were omitted, unfortunately. Most of citation are local. Reference list does not meet MDPI requirements
More comments are indicated in the text

Author Response

(The authors gave the same response as above.)

Reviewer 3 Report
Comments and Suggestions for Authors
The paper is mostly well written, though there are quite a few sentences errors which can make the meaning of some of the sentences unclear. The analysis appears to be rigorous but needs more detailed explanation in parts (especially in methods section). The discussion I also found confusing in parts.
ABSTRACT
LINES 26-27 COMPLEMENT THE JUSTIFICATION OF THE STUDY
LINES 26-30 THERE IS NO EXPLANATION OF THE METHODOLOGY??
LINE 39 CHECK WORD SPACING
KEYWORDS ADD CONSERVATION OR PRESERVING
INTRODUCTION
LINE 48 UPDATE REFERENCE
LINE 51 THE SCIENTIFIC NAME IS IN ITALIC
LINES 60-62 BRIEFLY JUSTIFY WHY THOSE RESERVATIONS WERE ESTABLISHED
LINE 71 EX-SITU IS IN ITALIC
LINE 84 REDUCE THE NUMBER OF REFERENCES, LEAVE THE MOST IMPORTANT ONE
LINE 95 REDUCE THE NUMBER OF REFERENCES, LEAVE THE MOST IMPORTANT ONE
MATERIALS AND METHODS
LINE 114 MENTION IN WHICH PART OF CHINA THIS PROVINCE IS LOCATED (E.G., CENTRAL CHINA)
FIGURE 1 COMPLEMENT THE INFORMATION IN THE FIGURE CAPTION, IT IS VERY SHORT
LINES 151-152 NEED A REFERENCE
LINE 159 CHECK WORD SPACING
TABLE 2 DELETE "THE" IN COLUMN 3 AND 4
RESULTS
FIGURE 2 IMPROVE RESOLUTION
FIGURE 3 COMPLEMENT THE INFORMATION IN THE FIGURE CAPTION, IT IS VERY SHORT
TABLE 3 CHANGE GE-NETIC BY GENETIC
TABLE 3 CHANGE POP-ULA-TIONS BY POPULATIONS
LINE 237 THIS IS VERY CONFUSING 2.4% ~52.5%; AND THIS IS REPEATED IN THE LINES 303, 312, 313, 322, 327, 329, 335
FIGURE 4 IMPROVE RESOLUTION
DISCUSSION
LINE 352 CHANGE 60.71% BY 60.7%
LINES 444-458 ADD REFERENCES TO SUPPORT THE INFORMATION
REFERENCES
REFERENCES DO NOT FOLLOW THE JOURNAL EDITORIAL STANDARDS
REFERENCES ARE PRESENTED WITHOUT DOI
LINE 499 CHANGE Cambridge university press by Cambridge University Press
LINE 524 change iucn by IUCN
LINE 524 change Iucnredlist.org by iucnredlist.org
SOME REFERENCES APPEAR WITH THE CITY AND OTHERS DO NOT; SOME REFERENCES APPEAR WITH THE USE OF & AND OTHERS WITH THE USE OF AND... IT MUST BE HOMOGENIZED
LINE 618 CHANGE ecology by Ecology
Comments on the Quality of English Language
Minor editing of English language required
Author Response

(The authors gave the same response as above.)

Round 2
Reviewer 2 Report
Comments and Suggestions for Authors
Authors have improved the MS considerably; however, some terms are less suitable for the translocation process of wildlife species (see comments in the text). Authors should be more careful as due to typos as to using of terms

Minor editing of English language required.
Author Response
Dear reviewer:
Thank you very much for taking the time to review our manuscript again. Based on your suggestions, we have reviewed the entire text and made revisions. We would like to express our gratitude once again for your efforts in revising this manuscript. We hope everything goes smoothly for you!
sincerely,
Jindong Zhang

Reviewer 3 Report
Comments and Suggestions for Authors
The authors have added and responded to all comments made to the manuscript, which has improved its content and quality. Therefore, I have no further comments or suggestions for the authors.
Author Response
Dear reviewer:
Thank you very much for taking the time to review our manuscript again. Your suggestions are very helpful in improving the quality of our article. We would like to thank you again for your affirmation of our article. We hope everything goes smoothly for you!
sincerely,
Jindong Zhang